# RE³: RETRIEVAL, RERANKING, AND REASONING FOR AGENTIC KNOWLEDGE GRAPH QUESTION ANSWERING

## ABSTRACT

Recently, the integration of Large Language Models (LLMs) and knowledge graphs has emerged as a promising approach for knowledge graph question answering by enhancing the reasoning capability in knowledge-intensive applications. However, existing methods face a key trade-off: they either introduce high computational costs when LLMs reason directly on graphs, or suffer from poor reasoning quality due to over-reliance on retrieval methods. To address this trade-off, we introduce a computationally efficient framework based on Retrieval, Reranking, and Reasoning (Re³). Specifically, we first develop "cognitively-informed retrieval" that improves subgraph retrieval quality via Question-Entity (Q-E) discrepancy scoring and hierarchical information aggregation. Second, we propose path-aware reranking, which employs lightweight cross-encoders to evaluate and prune reasoning paths efficiently. Last, we apply "agentic reasoning" to perform autonomous reasoning on high-quality subgraphs while balancing reasoning quality and computational overhead. Extensive experimental results on WebQSP and CWQ demonstrate that Re³ outperforms existing methods.

## 1 INTRODUCTION

Knowledge Graph Question Answering (KGQA) is a fundamental task in knowledge-intensive applications within Natural Language Processing (NLP), aiming to answer questions by reasoning over structured Knowledge Graphs (KGs) (Qiu et al., 2022). KGQA is widely used in open-domain QA, digital assistants, and scientific discovery systems (Talmor & Berant, 2018; Yih et al., 2016). However, it remains a very challenging task due to the need for precise graph traversal, contextual disambiguation, and multi-hop reasoning capabilities.

The recent development of Large Language Models (LLMs) significantly advanced this area, offering flexible and powerful reasoning capabilities (Brown et al., 2020). Consequently, LLMs have been increasingly incorporated into KGQA methods to enhance the flexibility of reasoning and reduce reliance on symbolic rules. However, LLMs still have inherent limitations in KGQA. For instance, LLMs are prone to hallucination, find it difficult to enforce structural constraints, and often lack access to complete knowledge (Zhao et al., 2023). Simply combining LLMs with KGs raises important questions about achieving optimal performance under resource constraints. How can we ensure both efficiency and accuracy? How can structured graph information be systematically leveraged during reasoning? How can we design modular components that can be flexibly combined based on resource constraints?

To address these challenges, recent studies focus on two main paradigms (Figure 1). First, **Agent-on-Graph** (Chen et al., 2024b; Ma et al., 2024; Ao et al., 2025) methods leverage LLMs' reasoning potential while lacking global information, necessitating frequent calls to advanced models lead to excessive computational costs, limiting practical applications. Second, **Retrieve-on-Graph** (Luo et al., 2024; Feng et al., 2024) methods retrieve relevant subgraphs as context. Despite their computational efficiency, they over-rely on retrieval methods and struggle to ensure the completeness and accuracy of retrieved content. The existing approaches lack a theoretical foundation grounded in cognitive science and information theory. Cognitive science suggests that reasoning integrates fast, global and intuitive judgments (retrieval) with slow, detailed deliberate analysis (reasoning) (Kahne-

Figure 1: A Comparison of Different KGQA Pipelines. Agent-on-Graph methods (a) often miss global context, whereas Retrieve-on-Graph methods (b) lack fine-grained details. In contrast, the Re$^3$ framework (c) provides a three-stage framework that integrates both.

man, 2011), while information theory emphasizes trade-off between information entropy and channel capacity (Miller, 1956).

Based on these insights from cognitive science and information theory, we hypothesize that the critical balancing capability for efficient and high-quality KGQA can be effectively modeled via a three-stage pipeline. Specifically, fast, cognitively-inspired retrieval ensures global context coverage; resource-constrained reranking optimizes information throughput; and agentic reasoning provides detailed, step-by-step analysis—together constituting our Re$^3$ framework. **Second**, we design a path-aware reranking module that aligns with the human brain's process for evaluating different reasoning paths. This stage applies a lightweight, fine-tuned cross-encoder model to assess semantic relevance and prune retrieved reasoning paths. It effectively removes noise while preserving the most pertinent paths organized into coherent reasoning chains. **Third**, we propose an agentic reasoning module that enables LLMs to perform autonomous reasoning on high-quality subgraphs constructed from the previous stages. The module can adaptively invoke graph query tools when information is insufficient, thereby reducing unnecessary LLM calls while improving performance on complex QA tasks.

Extensive experimental results obtained on WebQSP and CWQ demonstrate that Re$^3$ outperforms existing methods in complex multi-hop reasoning tasks. The main contributions of Re$^3$ include:

- **A theoretically principled KGQA framework:** We propose Re$^3$, a three-stage framework that systematically addresses the trade-off issue in existing methods. Re$^3$ achieves a favorable balance between reasoning accuracy and computational efficiency.

- **Advanced retrieval and pruning techniques:** We develop cognitively-informed retrieval and path-aware reranking strategies that significantly improve subgraph quality while maintaining high recall.

- **An agentic reasoning paradigm:** We implement a novel reasoning mechanism that enables LLMs to adaptively leverage graph query tools based on information sufficiency. This approach reduces computational overhead without sacrificing reasoning quality.

## 2 RELATED WORK

### 2.1 LLM-BASED KGQA

Early KGQA methods, which adopt semantic parsing (Berant et al., 2013) or information retrieval (Yao & Van Durme, 2014) techniques, often suffer from limited generalization, scalability issues, and poor handling of multi-hop or missing evidence queries (Du et al., 2023). In contrast, LLM-based approaches demonstrate promising results because they offer strong natural language understanding, flexible adaptation to varied queries, and support for in-context or chain-of-thought reasoning. Existing LLM-based KGQA methods can generally be categorized into "agent on graph" and "retrieve on graph" approaches.

The **agent on graph** methods enable LLMs to act as intelligent agents that actively explore knowledge graphs. Thinking-on-graph (Sun et al., 2024) and plan-on-graph (Chen et al., 2024b) utilize multi-step reasoning and incorporate reflection mechanisms. However, they require invoking the LLM at each reasoning step, resulting in a high computational cost when the graph size increases.

The **retrieve on graph** methods alleviate the reasoning burden on LLMs by pre-extracting relevant subgraphs as context. Representative works include SubgraphRAG (Feng et al., 2024) etc. They

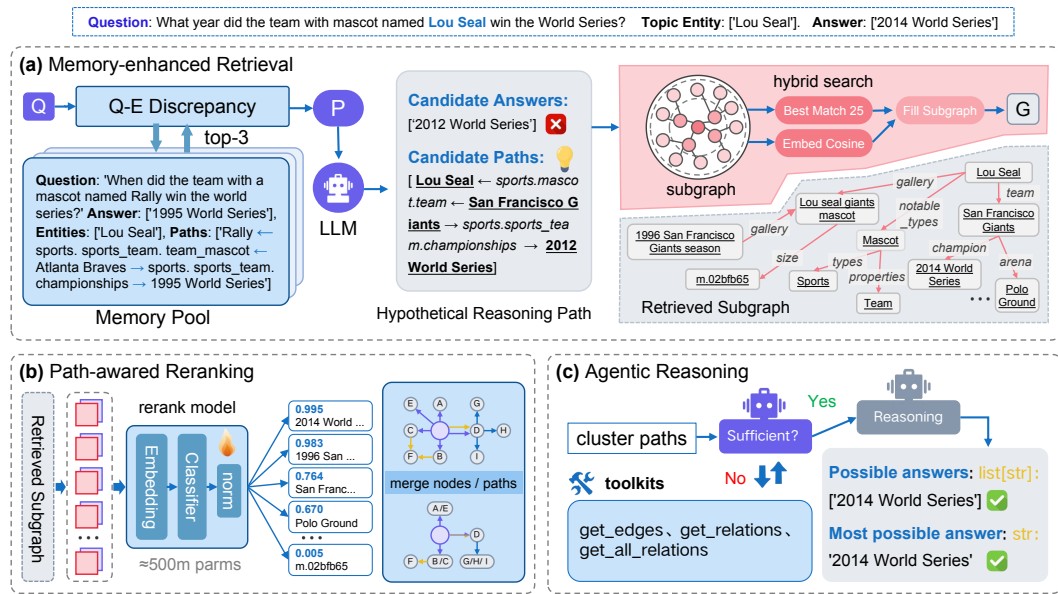

Figure 2: Re$^3$ has three main stages: (1) Cognitively-Informed Retrieval, which extracts high-quality subgraphs using Q-E discrepancy scoring and hierarchical aggregation; (2) Path-aware Reranking, which evaluates and prunes reasoning paths via a fine-tuned cross-encoder; and (3) Agentic Reasoning, which performs tool-augmented reasoning over refined subgraphs.

typically train specialized retrievers (e.g., MLPs or GNNs) to extract candidate subgraphs. However, these methods often over-rely on retriever performance and struggle to balance structural and semantic information, which can result in incomplete or imprecise contextual information.

## 2.2 AGENTIC RETRIEVAL-AUGMENTED GENERATION

Applying RAG to KGQA faces unique challenges. For example, entity relationship descriptions in KGs are often too concise and lack sufficient semantic context. Furthermore, it is difficult to both preserve and use structural information during retrieval. In addition, balancing retrieval efficiency and quality imposes significant computational constraints. Recent agentic RAG approaches (Yao et al., 2022; Yang et al., 2024) endow LLMs with autonomous decision-making and tool usage capabilities, enabling LLMs to select appropriate retrieval strategies based on the requirements of specific queries. However, integrating knowledge graphs into agentic systems requires careful agent architecture design to handle graph traversal complexity and reduce computational overhead during multi-hop reasoning.

The Re$^3$ framework proposed in this paper draws on insights from cognitive science and information theory, and achieves optimal integration of retrieval and reasoning through hierarchical information aggregation, semantic path evaluation, and agentic reasoning mechanisms. This design addresses the theoretical limitations of existing methods.

## 3 THE PROPOSED RE$^3$ METHOD

Given a Knowledge Graph (KG) $G = (E, R, T)$ and a natural language question $q$, where $E$, $R$, and $T$ denote the sets of entities, relations, and triples, the task of KGQA is to predict the answer entity set $A \subseteq E$ in response to $q$.

As illustrated in Figure 2, we formalizes the above problem into three stages: **cognitively-informed retrieval**, **path-aware reranking**, and **agentic reasoning**, which together refine reasoning quality while ensuring efficiency. Importantly, each module is designed as a plug-and-play component that can be independently enabled or disabled based on specific deployment requirements.

## 3.1 Cognitively-Informed Retrieval

This stage constructs a high-recall subgraph $S_q$ for each question $q$ via a retrieval function $\mathcal{R}(q, G, M)$, leveraging a cognitive memory pool $M$ to address the common problem of incomplete contexts in traditional question-entity retrieval.

We develop a two-stage retrieval approach that combines LLM-generated reasoning paths with a novel cognitively-informed mechanism. Our approach builds upon the observation that LLMs can generate plausible reasoning hypotheses, which, even if not entirely accurate, often align well with correct reasoning patterns.

First, inspired by Hypothetical Document Embeddings (HyDE) (Gao et al., 2023), we prompt LLMs to generate Hypothetical Reasoning Paths (HyRPs) which act as semantic bridges between questions and structured graph representations, offering richer retrieval signals than direct entity matching (Ji et al., 2024). Then, while HyRPs improve retrieval quality, they remain susceptible to LLM hallucinations and semantic drift. To mitigate these issues, we develop the cognitively-informed retrieval module that leverages a curated cognitive memory pool consisting of tra examples: *(Question, Topic Entity, Answers, Reasoning Paths)*.

Unlike conventional In-Context Learning (ICL) approaches (Brown et al., 2020) that often retrieve examples based on superficial similarity, our key innovation is the *Question-Entity (Q-E) Discrepancy* scoring mechanism. This mechanism explicitly favors examples with similar reasoning patterns (high question similarity) while discouraging excessive entity overlap. This ensures that retrieved examples match the logical core of the question, leading to higher-quality context for downstream reasoning.

For each question $q$ and the topic entity set $E$, we compute the Q-E Discrepancy score for memory candidates $m_i \in M$ based on cosine similarity:

$$\mathrm{D}(q, m_i) = \cos(q, m_i) - \frac{1}{|E|} \sum_{e_q \in E} \cos(e_q, m_i), \tag{1}$$

where $\cos$ denotes cosine similarity between the corresponding vector representations.

With enhanced HyRPs generated using in-context examples, we apply a hybrid retrieval strategy combining sparse methods (BM25 (Robertson et al., 2009)) for literal matches and a novel dense retrieval approach specifically designed for knowledge graph structures.

The cornerstone of our dense retrieval is a **hierarchical aggregation strategy** that captures both local relational semantics and high-level graph structure. This strategy implements a multi-layer ($L = 3$) information propagation framework. We first initialize entity and relation embeddings in complex space, where relations are modeled as rotations in complex space following RotatE (Sun et al., 2019). For any triple $(h, r, t)$, the embedding satisfies: $h^c \odot r^c \approx t^c$, where $\odot$ denotes element-wise complex multiplication. Then, the entity representations are iteratively refined through attention-weighted aggregation over neighbors:

$$h_e^{(\ell+1)c} = h_e^{(0)c} + h_e^{(\ell)c} + \sum_{e' \in \mathcal{N}(e)} \alpha_{e,e'}^{(\ell)} \left( h_{e'}^{(\ell)c} \odot \bar{r}_{e,e'}^c \right), \tag{2}$$

where $h_e^{(0)}$ is entity $e$'s initial embedding to avoid information loss, $h_e^{(\ell)}$ is its representation at layer $\ell$, and $\mathcal{N}(e)$ represents its neighbors. Attention uses the real part of the complex inner product with temperature and neighbor-wise softmax:

$$\alpha_{e,e'}^{(\ell)} = \mathrm{softmax}_{e' \in \mathcal{N}(e)} \Big( \mathrm{Re}(\langle h_e^{(\ell)c}, h_{e'}^{(\ell)c} \rangle) / \tau^{(\ell)} \Big). \tag{3}$$

Here $\langle x, y \rangle = \sum_k x_k \overline{y_k}$. Directionality and multi-relations: we treat the KG as directed and include inverse relations. For a neighbor $e'$, let $\mathcal{R}_{e,e'}$ be all relations between $e$ and $e'$ (including inverse), and $\mathcal{N}(e) = \{e' \mid |\mathcal{R}_{e,e'}| > 0\}$.

This hierarchical approach guarantees both semantic richness and structural completeness of subgraphs, thereby significantly improving retrieval accuracy and downstream reasoning quality.

## 3.2 PATH-AWARE RERANKING

Previous methods relying on dual-encoder architectures compute vector similarity between separately encoded question and path representations, ignoring fine-grained interactions between them, which leads to suboptimal path selection and redundant reasoning paths or subgraphs.

We propose a path-aware reranking module $P_q = \mathcal{K}(S_q, q)$, where $\mathcal{K}$ evaluates path relevance via fine-grained interaction modeling, outputting a pruned set $P_q$ of high-quality reasoning paths. To balance coverage and efficiency while avoiding excessive input length, we fine-tune a 500M-parameter reranker model to assign a confidence score to each question-path pair, reflecting how much the path semantically contributes to answering the question. The model adopts a cross-encoder architecture, where the concatenated question and path are jointly encoded by a pre-trained Transformer (Liu et al., 2019), followed by a feedforward classification head for scoring. Compared to dual-encoder approaches, cross-encoders enable full token-level interaction between the input pairs, which facilitates finer-grained semantic alignment and reasoning.

Moreover, due to the significant redundancy of tokens in the graph information represented as triples, as shown in Figure 2, we designed the "cluster" method to present graph information more efficiently while maintaining information density(e.g., sharing entities/relations).

Specifically, for each retrieved reasoning path, we group entities sharing identical relation types and topological positions relative to the question's topic entity. Each cluster is condensed into a *supernode* that aggregates entities with their shared relations, preserving semantic content and topological context for more compact LLM reasoning.

## 3.3 AGENTIC REASONING

Although recent agent-on-graph frameworks (Sun et al., 2024; Chen et al., 2024b), enhance reasoning by incorporating structured graph information, they have two key limitations. (1) Local-view bias: Limited access to global context and long-range dependencies causes LLMs to overlook critical entities, leading to suboptimal reasoning chains when processing noisy or non-central nodes. (2) Inefficient error correction: Existing approaches rely on simple backtracking mechanisms that are prone to getting stuck in loops or deviating from correct reasoning paths (Chen et al., 2024b).

To address the above limitations, we propose a subgraph-based agentic reasoning framework that enables LLMs with graph reasoning capabilities over refined subgraphs, expressed as $A = \mathcal{F}(P_q, q)$ where $\mathcal{F}$ is the reasoning function and $A$ is the predicted answer set. The LLM-driven agent dynamically decides to answer directly or invoke specific graph tools (entity/relation/path queries) when current information is insufficient. Once sufficient information is available, the LLM constructs a more coherent reasoning chain and outputs the final answer. Specifically, we provide the LLM with entity-query, relation-query, and path-query tools. The entity-query tool retrieves attributes and related relations based on entity names. The relation-query tool queries entity pairs that meet conditions based on relation types. The path-query tool queries possible reasoning paths based on starting entities and conditions.

This framework allows LLMs with autonomous decision-making abilities, enabling dynamic tool selection based on the current reasoning state. By enabling tool-aware, state-driven planning, our approach facilitates more flexible and adaptive multi-hop reasoning.

## 4 EXPERIMENTAL RESULTS

### 4.1 EXPERIMENTAL SETUP

**Datasets:** We evaluate our approach on two widely used multi-hop KGQA benchmarks: ComplexWebQuestions (CWQ) (Talmor & Berant, 2018), which contains 31,170 natural language questions, and WebQSP (Yih et al., 2016), which includes 4,454 questions over Freebase (Bollacker et al., 2008). Following RoG Luo et al. (2024), we use the Freebase snapshots released as `RoG-webqsp` and `RoG-cwq` on Hugging Face. Both datasets are challenging due to their requirements for accurate subgraph retrieval and complex reasoning.

Table 1: Main Results on Benchmark Datasets. The symbol − indicates that no performance was reported in previous publications. `None` indicates no fine-tuning was performed. (*-Path) denotes evaluation on refined datasets (WebQSP-Path, CWQ-Path; see Section 4.2)

| Method | LLM | Fine-tuned LLM | WebQSP | | CWQ | |
|---|---|---|---|---|---|---|
| | | | Hit | F1 | Hit | F1 |
| ToG (Sun et al., 2024) | GPT-4 | None | 82.6 | — | 69.5 | — |
| PoG (Chen et al., 2024b) | GPT-4 | None | 87.3 | — | 75.3 | — |
| RoG (Luo et al., 2024) | Llama2-7B | Llama2-7B | 85.7 | 70.8 | 62.6 | 56.2 |
| EPERM (Long et al., 2025) | Llama2-7B | Llama2-7B | 88.8 | 72.4 | 66.2 | 58.9 |
| GCR (Luo et al., 2025) | GPT-4o-mini | Llama-3.1-8B | 92.2 | 74.1 | 75.8 | 61.7 |
| SubgraphRAG (Feng et al., 2024) | GPT-4o-mini | None | 90.1 | 77.5 | 62.0 | 54.1 |
| FRAG (Guo et al., 2025) | GPT-4o-mini | None | 86.7 | — | 68.0 | — |
| Re$^3$ (*-Path) | GPT-4o-mini | None | 93.8 | 77.5 | 69.5 | 59.1 |
| Re$^3$ | GPT-4o-mini | None | 91.5 | 73.1 | 66.4 | 57.5 |

**Evaluation Metrics:** Following previous works (Sun et al., 2024; Feng et al., 2024), we adopt two standard metrics: Hit and F1 score for fair comparison with baselines. Hit measures if any correct answer appears in the results, and F1 score combines precision and recall, calculated as the macro-average across all questions. For internal analysis and ablation studies, we additionally use Hit@1 to evaluate if a correct answer is the top prediction.

**Implementation Details:** We employ GPT-4o-mini as the default base model. For memory retrieval, hyperparameters were selected empirically: we set top-$k$ to 3, apply pruning cut rates of 30% for initial retrieval and 20% for reranking, and use a pruning threshold of $t = 0.01$. The only component requiring additional training is BGE-reranker-v2-m3 (Chen et al., 2024a), which is fine-tuned on our self-constructed dataset (see Section 4.2) with a learning rate of 4e-5 for 10 epochs on an RTX 4090 GPU, taking approximately 40 minutes.

## 4.2 DATA PREPROCESSING AND RERANKER TRAINING

Prior KGQA research often uses shortest paths as gold standard reasoning paths (Luo et al., 2024; Feng et al., 2024), which often fail to capture the question's logical reasoning requirements. Shortest paths may connect entities through irrelevant relations that misalign with the question intent.

We address this by constructing valid reasoning paths as follows: extract all candidate paths of length at most 3 between topic entities and answer nodes, then use GPT-4o to rank and validate them based on semantic relevance and logical consistency with the question. And we further filter out questions from WebQSP whose answers are not present in the knowledge graph, as these cannot be solved through graph-based reasoning.

As a result, we obtained WebQSP-Path dataset (4,246 questions, 1,501 for testing) and CWQ-Path dataset (31,558 questions, 3,531 for testing). These validated reasoning paths serve two purposes: building the cognitive memory pool for our retrieval module and providing high-quality training data for the reranking component.

We construct reranker training datasets from WebQSP-Path and CWQ-Path via three strategies: (1) Adaptive construction, which modifies valid reasoning paths by adding, removing, or replacing sub-paths; (2) Heuristic traversal, which performs beam search from the topic entity to collect candidate paths; (3) Model-aware sampling, which leverages initial (non-finetuned) rerank scores to identify high- and low-confidence paths. We treat paths containing answers as positive examples, and sample hard negative examples from the remaining candidates to strengthen contrastive supervision.

We use the finetuned reranker model to compute confidence scores for all candidate paths from the topic entity to answer nodes, retaining the top 20% highest-scoring paths. We additionally filter candidates by a minimum score threshold of 0.01, yielding our final retrieved paths, which construct the final pruned subgraph. This results in a reranker training dataset containing 11,125 training samples and 4,132 test samples. Each sample includes a question with corresponding positive and negative path lists, averaging 7.8 positive and 46.2 negative paths per training sample. For test samples, we have an average of 3.2 positive and 29.2 negative paths per sample.

Table 2: Retrieval Performance with Different Strategies.

| Strategy | $ER_{retrieve}$ | $TR_{retrieve}$ | $Hit_{kgqa}$ | $F1_{kgqa}$ |
|---|---|---|---|---|
| $Re^3$ | 99.9 | 93.5 | 93.8 | 77.0 |
| w/o Memory | 98.5 | 90.2 | 92.4 | 73.2 |
| w/o Agg | 97.3 | 89.5 | 92.8 | 73.9 |

Table 3: Ablation Study on The WebQSP-Path across Different Hops (Hit only).

| Strategy | 1-hop | 2-hop | Full |
|---|---|---|---|
| $Re^3$ | **95.1** | **92.8** | **93.8** |
| w/o Reranking | 93.6 | 88.6 | 91.9 |
| w/o Agentic | 91.8 | 92.0 | 91.8 |

Note that, when reporting experimental results, we use the original WebQSP and CWQ testing sets by default for fair comparison with existing methods, while selected ablation studies use the refined datasets for isolating the impact of reasoning path quality.

## 4.3 PERFORMANCE COMPARISON

We compare $Re^3$ against the following baseline methods: (1) Agent-on-Graph approaches: ToG (Sun et al., 2024) and EPERM (Long et al., 2025); (2) Retrieval-on-Graph approaches: RoG (Luo et al., 2024), GCR (Luo et al., 2025), and SubgraphRAG (Feng et al., 2024).

Table 1 presents performance comparisons on WebQSP and CWQ. $Re^3$ achieves consistently strong performance across standard evaluation metrics. On WebQSP, our framework achieves 91.5% Hit and 73.1% F1, while on CWQ, it attains 66.4% Hit and 57.5% F1. These high Hit values indicate that $Re^3$ can consistently identify correct answers within its predictions.

$Re^3$'s superior performance stems from its synergistic combination of global and fine-grained retrieval. The cognitively-informed retrieval provides comprehensive global context through hierarchical information aggregation, while the agentic reasoning enables targeted exploration of specific graph regions when needed.

## 4.4 ABLATION STUDIES

We conduct comprehensive ablation studies on the WebQSP-Path dataset to verify the contribution of each component in our $Re^3$ framework. These studies systematically demonstrate how each module contributes to the overall performance and validate our core design principles. The plug-and-play nature of our components allows for flexible configuration, as evidenced by the independent evaluation of each module's impact.

**Impact of Cognitively-Informed Retrieval:** To evaluate the effectiveness of the cognitively-informed retrieval module, we design experiments comparing different memory configurations. Table 2 presents results where "w/o Memory" removes the memory pool entirely, relying solely on question-based retrieval.

Furthermore, leveraging the refined WebQSP-Path dataset, we use $ER_{retrieve}$ to denote the recall of answer entities in the retrieved subgraph and $TR_{retrieve}$ to denote the recall of reasoning-relevant triples (i.e., triples that directly contribute to answering the question). And we use $_{kgqa}$ to denote metrics for the question-answering task,

The results reveal several key insights. First, removing the cognitive memory pool (w/o Memory) results in a 3.8% decrease in $F1_{kgqa}$, from 77.0% to 73.2%, indicating that contextual memory significantly enhances retrieval quality. The $ER_{retrieve}$ decreases from 99.9% to 98.5%, demonstrating that memory assists both in identifying relevant information and maintaining comprehensive coverage.

These results validate our core hypothesis that cognitively-informed retrieval provides essential contextual guidance for subgraph extraction. The modular design allows this component to be easily integrated or removed based on system requirements.

**Impact of Hierarchical Aggregation Strategy:** We conducted ablation studies on the aggregation module using a simple knowledge graph embedding (KGE) method to represent the vector graph. As shown in Table 3, when maintaining the same context window size, the "w/o Agg" variant achieves a 73.9% hit rate on the KGQA task, compared to 77.0% for the full model. Specifically, $ER_{retrieve}$ decreased from 99.9% to 97.3%, and the hierarchical aggregation strategy achieved $TR_{retrieve}$ of 93.5%, significantly outperforming the 89.5% achieved by the simple KGE method.

Table 4: Performance of Reranking Models

| Category | Method | $F1_{rerank}$ | $Hit@1_{kgqa}$ | $F1_{kgqa}$ |
|---|---|---|---|---|
| Original | bge-reranker-v2-m3 | 19.9 | 80.8 | 69.3 |
| | qwen3-reranker-0.6b | 23.3 | 80.4 | 68.9 |
| Fine-tuned | reranker (adaptive-only) | 87.1 | 83.3 | 75.9 |
| | reranker (heuristic-only) | 86.0 | 84.6 | 75.4 |
| | **reranker (all)** | **88.6** | **86.7** | **77.5** |

**Impact of Reranking Model:** In WebQSP dataset testing, retrieved subgraphs $S_q$ initially contained 930 nodes on average, reduced to 42 nodes after reranking. As Table 3 shows, removing the reranking module caused the Hit metric to drop from 93.8% to 91.9% while increasing average LLM input token consumption from 4,028 to 14,765. Therefore, the reranking module represents an optimal trade-off between accuracy and efficiency, simultaneously reducing computational costs and enhancing model performance by focusing on the most relevant knowledge graph components. This component can be seamlessly integrated or bypassed depending on computational constraints.

Table 4 presents the effects of different reranking strategies, comparing both original and fine-tuned reranking models. $F1_{rerank}$ represents the path-level F1 score evaluated on the reranker's test set; $Hit@1_{kgqa}$ and $F1_{kgqa}$ represent performance metrics on the final KGQA task after all retrieval, reranking, and reasoning steps (note: Hit@1 is used here for internal analysis).

The results show that fine-tuned reranking models improve retrieved path quality. For instance, the fine-tuned model trained on the combined dataset (including adaptive, heuristic, and model-aware samples) achieves 88.6% $F1_{rerank}$ on the reranking test set, in contrast to 19.9% of the original model. More importantly, this improvement translates to better KGQA performance, with $Hit@1_{kgqa}$ increasing from 80.8% to 86.7% and $F1_{kgqa}$ improving from 69.3% to 77.5%.

**Impact of Agentic Reasoning:** To verify the effectiveness of agentic reasoning, we compare $Re^3$ with $Re^3$ w/o `Agentic` (removing agentic reasoning) across different question complexity scenarios. Table 3 reports performance under varying hops in the dataset.

The results demonstrate that removing the agentic reasoning component leads to a notable performance degradation, with overall Hit rate decreasing by 2.0 percentage points (from 93.8% to 91.8%). We observe that this performance gap varies by question complexity: in 1-hop scenarios, performance drops by 3.3 percentage points (from 95.1% to 91.8%), while in 2-hop scenarios, it decreases by only 0.8 percentage points (from 92.8% to 92.0%).

This pattern suggests that for some 1-hop questions, initially retrieved context is insufficient due to ambiguities or missing links; agentic reasoning helps dynamically supplement missing facts for these cases. This dynamic supplementation of reasoning chains enables $Re^3$ to resolve uncertainties that static retrieval-only approaches cannot address. These findings validate our hypothesis that integrating agentic reasoning capabilities significantly enhances model performance by allowing dynamic, context-aware information gathering during the reasoning process.

## 4.5 CROSS-MODEL GENERALIZATION

Table 5 demonstrates $Re^3$'s comprehensive performance across different LLMs, examining both effectiveness (Hit, Hit@1) and reliability (Hall@1, Hall).

To ensure transparency in our hallucination analysis, we formalize our detection protocol: a hallucinated answer is defined as any predicted answer entity that does not exist in the target knowledge graph (KG) used for evaluation. Even if such an entity may correspond to a factually correct real-world answer, we mark it as hallucinated because it is not grounded in the provided, reliable information source (the KG). Here, Hall@1 measures the percentage of questions where the top-1 answer contains hallucinated content, while Hall captures the overall percentage of hallucinated content across all generated answers. To ensure fair comparison, we conducted all experiments without using thinking mode for all models.

The results demonstrate that $Re^3$ maintains consistent performance across varying model scales and architectures, with Hit rates of 89.2–94.2% and Hit@1 rates of 82.5–88.6%, indicating strong architecture-agnostic generalization. Among closed-source models, GPT-4.1 achieves the highest

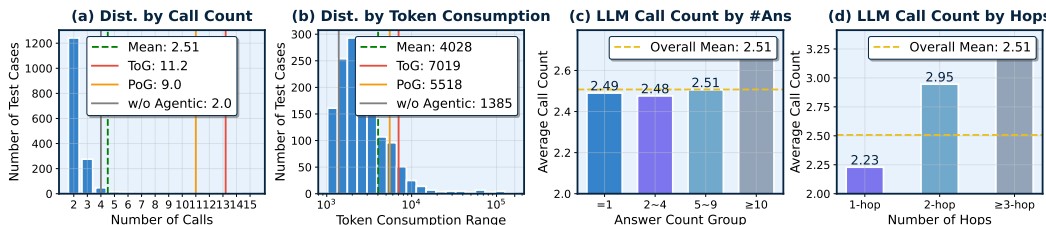

Figure 3: Analysis of Re[3] in Terms of Average Token Consumption across Different Models

Table 5: Performance and Hallucination Analysis of Re[3] across LLM Models on WebQSP-Path

| Category | Base Model | Hit↑ | Hit@1↑ | Hall↓ | Hall@1↓ |
|---|---|---|---|---|---|
| Closed-source | Re[3]+GPT-4o | 93.8 | 86.9 | 10.39 | 5.06 |
| | Re[3]+GPT-4.1 | 94.2 | 88.6 | 9.06 | 5.53 |
| | Re[3]+Gemini-2.5-Pro | 93.8 | 85.3 | 14.79 | 7.46 |
| Open-source | Re[3]+DeepSeek-V3 | 91.3 | 87.4 | 7.88 | 3.94 |
| | Re[3]+Qwen3-235B-A22B | 91.5 | 86.7 | 6.66 | 3.13 |
| Configuration | Re[3] w/o Agentic | 91.9 | 84.2 | 15.12 | 8.86 |
| | Re[3] (full) | 93.8 | 86.7 | 9.93 | 4.26 |

accuracy (Hit: 94.2%, Hit@1: 88.6%), while among open-source models, DeepSeek-V3 exhibits exceptional performance with the highest Hit@1 (87.4%) and the lowest hallucination rates (Hall@1: 3.94%, Hall: 7.88%).

The configuration comparison reveals the critical role of our agentic reasoning component in mitigating hallucinations. The Re[3] framework reduces Hall@1 by 51.9% (from 8.86% to 4.26%) and Hall by 34.3% (from 15.12% to 9.93%) compared to the variant without agentic reasoning. This substantial improvement validates our hypothesis that autonomous verification mechanisms significantly enhances factual groundedness by enabling models to verify answers dynamically against the knowledge graph during reasoning.

### 4.6 EFFICIENCY ANALYSIS

The efficiency analysis of Re[3] reveals insights into the framework's computational costs. As illustrated in Figure 3, the average token consumption increases notably when the reranking module is removed, indicating that the reranking process effectively reduces unnecessary token usage by filtering out irrelevant information.

This efficiency is crucial in multi-hop scenarios, where the complexity of questions often leads to increased token consumption. The ability to dynamically supplement reasoning chains with targeted information retrieval enables Re[3] to achieve a better trade-off between performance and resource utilization than retrieval-only approaches.

### 5 CONCLUSION

This paper presents Re[3] (Retrieval-Reranking-Reasoning), a theoretically principled framework that addresses fundamental trade-offs in KGQA. These components synergistically realize our theoretical framework's objective: combining the comprehensive coverage of retrieval methods with the precise reasoning capabilities of agent methods, while circumventing the inherent limitations of both paradigms.Extensive experiments on WebQSP and CWQ demonstrate that Re[3] outperforms existing methods, particularly excelling in complex multi-hop reasoning while substantially reducing computational overhead.

In our engineering practice, the Re[3] framework can be seamlessly integrated into current multi-agent systems as a sub-agent for agentic knowledge graph retrieval. Future work will explore the construction of a question answering paradigm over temporal knowledge graphs, aiming to better integrate KGs with LLMs.

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

# A  DATASETS

We evaluate our approach on two complex multi-hop KGQA datasets: ComplexWebQuestions (CWQ) (Talmor & Berant, 2018) and WebQSP (Yih et al., 2016). The detailed dataset statistics are presented in Table 6. Both WebQSP and CWQ utilize Freebase KGs (Bollacker et al., 2008) for reasoning and contain multi-hop, multi-answer complex question answering instances.

Table 6: Dataset Statistics

| Statistic | WebQSP | CWQ |
|---|---|---|
| #Train Samples | 2,826 | 27,639 |
| #Test Samples | 1,628 | 3,531 |
| %Answers = 1 | 51.2% | 70.6% |
| %Answers 2–4 | 27.4% | 19.4% |
| %Answers 5–9 | 8.3% | 6.0% |
| %Answers ≥10 | 12.1% | 4.0% |
| %1-hop Questions | 65.5% | 40.9% |
| %2-hop Questions | 34.5% | 38.3% |
| %≥3-hop Questions | 0.0% | 20.8% |

# B  REPRODUCIBILITY STATEMENT

## B.1  ABLATION STUDIES

We conducted several ablation experiments with the following configurations:

`w/o Memory`: We eliminated all components related to the memory pool and HyPRs design, instead utilizing only the question and entities directly for subgraph retrieval.

`w/o Agg`: We ablated the hierarchical aggregation strategy entirely, substituting it with a direct application of dense semantic vectors from nodes and edges as knowledge graph embeddings.

`w/o Reranking`: We employed a composite retrieval scoring mechanism that incorporated both BM25 scores and semantic similarity between questions and reasoning paths, retaining 50% of nodes based on this metric.

`w/o Agentic`: In this configuration, we bypassed the judgment and retrieval processes, instead feeding the question and path clusters directly to the LLM for answer generation.

## B.2  THE SNAPSHOT FOR LLMS

Here, we report the specific version numbers of the different LLMs used in the experiments, along with the available parameter counts.

Table 7: Specific Version and Parameter Information of LLMs Used in Re[3]

| Version | Params |
|---|---|
| gpt-4o-2024-11-20 | *Not disclosed* |
| gpt-4o-mini-2024-07-18 | *Not disclosed* |
| gpt-4.1-2025-04-14 | *Not disclosed* |
| gemini-2.5-pro-preview-06-05 | *Not disclosed* |
| deepseek-v3-250324 | 670B / 37B (MoE) |
| qwen3-235b-a22b-instruct-2507 | 235B / 22B (MoE) |

## B.3 INITIAL PROMPT (WITH IN-CONTEXT SAMPLES)

```
Initial Prompt Template

You are a helpful assistant designed to output JSON that aids in
navigating a knowledge graph to answer a provided question.
The response should include the following keys:

(1) reasoning_paths:  List[str], a list of reasoning steps that
should be used to answer the question.
(2) candidate_answers:  List[str], a list of candidate answers that
may be used to answer the question, answers should be simple and
specific as an Entity.

{samples}

Q: {question}, {q_entity}
A:
```

where {question} denotes the natural language question, {q_entity} the topic entity set, and {samples} in-context examples.

## B.4 FINAL AGENT PROMPT (WITH SAMPLES AND TOOL-USE)

```
Final Agent Prompt Template

You are a helpful assistant designed to output JSON that aids
in using the provided question and the reasoning paths from a
knowledge graph,
you can think about the question and the reasoning paths carefully,
and give the answer (a node) based on the reasoning paths and your
knowledge.

REMEMBER:
(1) ONLY when you cannot find possible answers based on those
paths, you can use the tools to find the possible answers.
(2) If the tools cannot find the possible answers, you can use your
knowledge to find the possible answers.
(3) You have to give the possible answers based on the previous
tools result, reasoning paths and your knowledge.
(4) IF the answer ENTITY is in the context(tools result, reasoning
paths), you have to give the answer the same as the context.

The JSON format response should include the following keys:
(1) possible_answers:  List[str], all possible answers to the
question.
(2) most_possible_answer:  str, the most possible answer to the
question (CANNOT be empty).

Here are some examples of the question and the reasoning paths:

{samples}

Reasoning paths:  {reasoning_paths}
Question:  {question}

Answer:
```

where {reasoning_paths} denotes reranked paths fed to the agent, {question} the input question, and {samples} few-shot examples.

## B.5 KG TOOL PSEUDOCODE

Regarding tool invocation, we primarily rely on the LLM's built-in function-calling capability. In the request's parameter configuration, we specify the functions' descriptions, parameters, and return values as a JSON specification, with detailed definitions. The LLM can autonomously select functions to invoke, analyze the execution results, and plan subsequent actions.

We present pseudocode specifications for the major KG querying tools used in our agentic framework. Each tool is illustrated with its typical SPARQL implementation.

Specific implementation details can be found in our publicly available code repository at: https://anonymous.4open.science/r/Re3-0FFE

---

**entity_query(e)**

*Purpose:* Given an entity $e$, retrieve all relations (predicates) associated with $e$.
*Pseudocode:*

```
Input: e (entity)
SPARQL: SELECT ?r WHERE {
            e ?r o
        }
```

---

**triple_query(e, r)**

*Purpose:* Given an entity $e$ and a relation $r$, retrieve all objects $o$ such that $(e, r, o)$ holds.
*Pseudocode:*

```
Input: e (entity), r (relation)
SPARQL: SELECT ?o WHERE {
            e r ?o
        }
```

---

**entity_relation_type(e)**

*Purpose:* Given an entity $e$, retrieve the types of all relations associated with $e$
*Pseudocode:*

```
Input: e (entity)
SPARQL: SELECT DISTINCT ?type WHERE {
            e ?r ?o .
            ?r rdf:type ?type
        }
```

---

**triple_relation_type(e, r)**

*Purpose:* Given an entity $e$ and relation $r$, return the type of relation $r$ for $e$.
*Pseudocode:*

```
Input: e (entity), r (relation)
SPARQL: SELECT DISTINCT ?type WHERE {
            e r ?o .
            r rdf:type ?type
        }
```

---

## B.6 Valid Path Selection Prompt

> **Valid Path Selection Prompt Template**
>
> ```
> Seriously analyze these paths, which of these paths can be used to
> answer this question:
> <Question>{question}</Question>
> <Answer_Entity>{answer_entity}</Answer_Entity>
>
> NOTE: Please response with the following format:
> Thought: <your thought>
> Answer: <path id>
>
> The paths are: {path_str}
> Response:
> ```

where {question} denotes the input question, {answer_entity} the gold answer entity set, and {path_str} the enumerated candidate paths with indices.

# C Detailed Measurement

## C.1 Hallucination Detection Protocol

We formalize our detection protocol for hallucinated answers in KGQA.

**Definition.** We define a hallucinated answer as any predicted answer entity that does *not* exist in the target knowledge graph (KG) used for evaluation (Freebase snapshot aligned with each dataset). Even if such an entity may correspond to a factually correct real-world answer, we still mark it as hallucinated because it is not grounded in the provided, reliable information source (the KG).

**Metrics: Hall@1**: the percentage of questions whose top-1 predicted answer is hallucinated (i.e., the top-1 prediction cannot be resolved to a KG entity).

**Metrics: Hall**: the percentage of evaluation instances whose predicted answer set contains any hallucinated entity. For each instance, if any predicted entity is out-of-KG, it is counted as 1; otherwise 0. Hall is the mean over instances (akin to a hit-style computation at the instance level).

## C.2 Efficiency Measurement Protocol and Cross-Paper Comparisons

Here, we clarify the data sources and our accounting protocol.

**Data sources for baselines.** The numbers for the compared methods (e.g., number of model calls, token counts) are *directly taken* from their publicly available papers, appendices, or official repositories. We did not re-implement or re-measure these systems; thus their efficiency figures are reported "as published." When a prior work provides multiple metrics or variants, we follow the authors' primary setting.

**Our accounting policy.** All token counts reported for Re$^3$ are *all-inclusive* and computed end-to-end per query. Specifically, we count: (i) all prompt tokens (system/developer/user), (ii) all tool-use messages including function/tool-call arguments and tool outputs when fed back to the model, (iii) agent/controller messages, retries, and rollbacks, and (iv) model outputs. We also include tokens associated with tool interactions even when cached results are returned (i.e., cache hits are *not* excluded), to reflect the actual message traffic processed by the model.

**Tokenizer and budgets.** We use the official tokenizer associated with each model provider and count both input and output tokens. Within our own experiments, prompts and budgets are held fixed across models unless otherwise noted.

**Interpretation and limitations.** Because external works may differ in tokenizer choice, inclusion/exclusion of tool messages, prompt templates, and budget constraints, cross-paper comparisons can only be approximate. However, our conservative, all-inclusive accounting for Re$^3$ imposes a stricter measurement protocol, tending to slightly over-count rather than under-count. This makes

our efficiency numbers comparable under a uniform and conservative policy, while baseline numbers are faithfully reproduced from the literature without modification.

