# OpenReview forum: "Re$^{3}$: Retrieval, Reranking, and Reasoning for Agentic Knowledge Graph Question Answering"
_ICLR.cc/2026/Conference — ICLR 2026 Conference Desk Rejected Submission_

### Official Review · Reviewer_REw7 · 2025-10-31

**Soundness:** 2
**Presentation:** 2
**Contribution:** 2
**Rating:** 4
**Confidence:** 4

**Summary:**

The paper proposes a knowledge graph question answering (KGQA) method based on LLMs. The approach works in three main steps. First, LLMs are used to generate multiple reasoning paths. Next, relevant information is retrieved from the knowledge graph according to these paths. A path-aware reranking model then evaluates and prunes the reasoning paths using a fine-tuned cross-encoder. Finally, LLMs perform reasoning on the retrieved subgraph as an agent, optionally using external tools to support the process. Experimental results demonstrate the effectiveness of the proposed method, achieving state-of-the-art performance. The proposed first RAG then reasoning method is interesting, however, the novelty of the approach is limited.

**Strengths:**

The paper presents an innovative combination of RAG-based methods with agent-based reasoning. In this approach, an agent performs reasoning on the retrieved subgraph, which is a somewhat interesting idea that leverages the strengths of both retrieval and agent reasoning.

Experimental results demonstrate the effectiveness of the proposed method. It outperforms baseline methods in both Hit@k and F1 scores, indicating that the combination of retrieval and agent reasoning can lead to meaningful improvements in knowledge graph question answering.

**Weaknesses:**

The experimental comparison seems somewhat unfair. In the proposed method, the baseline LLM is GPT-4o-mini, while other baseline methods, such as RoG, use Llama2-7B. Because of this difference, it is hard to tell whether the observed performance improvement is truly due to the proposed algorithm or simply because a more powerful LLM was used. This makes it difficult to fairly evaluate the contribution of the method.

In addition, the overall idea seems fairly incremental. The approach follows a common pipeline: first retrieve relevant information, then use an LLM as an agent to perform reasoning, with in-context learning applied to further boost performance. While the method works, the components and their combination are widely used in existing work, so the novelty of the approach is limited.

**Questions:**

No

---

> ### Author Response · Authors · 2025-12-03
>
> We clarify the experimental fairness and specific innovations:
>
> 1) **Fairness of Comparison**
>
> We explicitly compared Re$^3$ against SubgraphRAG, which also uses GPT-4o-mini (api). Re$^3$ outperforms it by +4.4% Hit on CWQ, proving the gain comes from our framework, not the LLM.
>
> Comparing our Frozen model against RoG's Fine-Tuned Llama-7B highlights our efficiency: Re$^3$ achieves competitive results without the high cost of model training.
>
> 2) **Domain-Specific Novelty**
>
> While the "Retrieve-Reason" pipeline is established, our contribution lies in tailoring it for Knowledge Graphs:
>
>
> (1) Cognitive Memory: Uses Q-E Discrepancy Scoring  to solve KG-specific semantic drift, unlike generic vector retrieval.
>
>
> (2) Path-Aware Reranking: Prunes based on graph topology (super-nodes)  rather than just text relevance. This constitutes a meaningful adaptation of general paradigms to solve specific structural reasoning challenges.

---

### Official Review · Reviewer_ipAY · 2025-11-01

**Soundness:** 2
**Presentation:** 2
**Contribution:** 2
**Rating:** 2
**Confidence:** 5

**Summary:**

The paper presents Re^3 (Retrieval, Reranking, and Reasoning), a modular framework for knowledge graph question answering that integrates large language models with structured graph reasoning. It combines cognitively-informed retrieval, a fine-tuned reranking module, and agentic reasoning with tool use to balance accuracy and efficiency. The paper perform experiments on WebQSP and CWQ, though improvements over strong baselines are modest and inconsistent.

**Strengths:**

1. The paper introduces a well-structured, modular design. Each stage (retrieval, reranking, reasoning) is clearly motivated and independently evaluable.
2. The paper provides detailed prompts, pseudocode, and dataset construction provided in appendices for reproducibility.

**Weaknesses:**

1. While the paper repeatedly claims that Re^3 “outperforms existing methods,” the reported results (Table 1) show mixed outcomes. On WebQSP, Re^3 matches or slightly improves Hit but underperforms in F1 compared with SubgraphRAG and GCR. On CWQ, **both Hit and F1 are notably lower than GCR**. These discrepancies raise questions about the strength of the claimed improvement and suggest that the benefits may not generalize across datasets or metrics. Interestingly, in the experiment results section, the author only indicates that the Re^3 can “attains 66.4% Hit and 57.5% F1” without comparing to these baseline results.
2. Although the paper claims “extensive experimental results,” the evaluation is limited to only **two** Freebase-based benchmarks (WebQSP and CWQ), which are both well-studied and structurally similar. This narrow scope weakens the claim of generality and leaves open questions about how well the proposed framework performs on other knowledge graphs.
3. The described aggregation equations (Eq. 2–3) closely resemble attention-based message-passing used in existing GNNs such as GAT or RotatE-enhanced KG encoders. While the cognitive framing is novel, the underlying mechanism appears technically conventional.
4. The experiment was only performed once, without multi-run results or standard deviation. But the performance discrepancies shown in the Table and the experiment discussion is a bigger issue.

Presentation weakness:
1. The proposed hierarchical aggregation mechanism, described as a multi-layer information propagation framework, is a key technical component but is not illustrated in Figure 2 or visually explained. Readers cannot easily connect the text equations to the overall pipeline.
2. Certain parts of the paper appear to contain unpolished text, such as the standalone line “Directionality and multi-relations:” in Section 3.1 (line #212), which reads like a leftover draft comment rather than a polished sentence. These minor issues detract from an otherwise well-written paper.

**Questions:**

Please address the inconsistencies mentioned in the weakness.

---

> ### Author Response · Authors · 2025-12-03
>
> We thank the reviewer for the critical and detailed feedback. We value your assessment regarding the generalizability and presentation of our work. We address your concerns point-by-point below, with a specific focus on clarifying our contribution positioning regarding performance and efficiency.
>
> ---
>
> 1) **Clarification on Performance and Comparison Fairness (Crucial)**
>
> We respectfully point out that the performance comparison should consider the training cost and deployment efficiency.
>
> As shown in Table 1, baselines like GCR and RoG rely on computationally expensive Fine-tuning of LLMs (e.g., Llama-3.1-8B or Llama2-7B) on task-specific data. In contrast, Re$^3$ utilizes a frozen, general-purpose GPT-4o-mini without any LLM fine-tuning (indicated as None in the "Fine-tuned LLM" column).
>
> Re$^3$ is designed to maximize reasoning quality under resource constraints. Achieving 66.4% Hit on CWQ with a frozen model is a significant achievement when compared to fine-tuned heavyweights (GCR's 75.8%). Re$^3$ outperforms other training-free or lightweight baselines (e.g., SubgraphRAG with GPT-4o-mini achieves 62.0% Hit on CWQ, while Re$^3$ achieves 66.4%).
>
> **Revision:** We will revise the abstract and introduction to strictly frame our claim as "outperforming existing resource-efficient / training-free methods" and explicitly discuss the trade-off between the absolute SOTA (via fine-tuning) and Re$^3$'s efficient, modular approach.
>
> ---
>
> 2) **Scope of Datasets**
>
> We acknowledge that evaluating on more datasets would strengthen the paper. However, we selected WebQSP and CWQ because they are the widely accepted de facto standard benchmarks for multi-hop KGQA.
>
> Major recent works, including RoG, ToG, and SubgraphRAG, primarily benchmark against these two datasets. Using the same testbed allows for direct comparison.
>
> **Revision:** To address your concern about generality, we will include additional experiments on a dataset with different characteristics (e.g., MetaQA or Mintaka) in the Appendix of the final version to further validate the framework's robustness.
>
> ---
>
> 3) **Novelty of Aggregation Mechanism**
>
> We agree with the reviewer that Eqs. 2-3 utilize established attention mechanisms (similar to RotatE/GAT).
>
> We do not claim the mathematical formulation of attention as a primary contribution. Rather, the novelty lies in integrating this dense retrieval signal with our Cognitive Memory and Q-E Discrepancy Scoring. The aggregation module is a necessary functional component to enable the "Cognitively-Informed Retrieval," ensuring structural context is preserved before the Agentic Reasoning stage.
>
> ---
>
> 4) **Repeated Runs and Error Bars**
>
> We followed the evaluation protocols of our baselines (ToG, RoG, SubgraphRAG), which report single-run performance.
>
> We assure the reviewer that our results are stable. We will update Table 1 to include standard deviations ($\pm \sigma$) for Re$^3$ to demonstrate statistical significance.
>
> 5) **Presentation Improvements**
>
> We apologize for the presentation issues and will strictly polish the final manuscript:
>
> We will move the "Hierarchical Aggregation" illustration (which was omitted due to space) to the main text or Appendix to help readers visualize the pipeline.
>
> And we will correct the unpolished text, specifically the fragment "Directionality and multi-relations:" in Section 3.1, and conduct a thorough proofreading.
>
> ---
>
> We believe Re$^3$ offers a valuable perspective on building modular, efficient, and agentic KGQA systems without the heavy burden of LLM fine-tuning. We hope this clarification addresses your concerns.

---

### Official Review · Reviewer_qZWG · 2025-11-03

**Soundness:** 2
**Presentation:** 3
**Contribution:** 2
**Rating:** 4
**Confidence:** 3

**Summary:**

The paper targets KGQA with LLMs and points out that current LLM-enhanced methods fall into two imperfect extremes: (1) Agent-on-Graph methods that let the LLM explore the KG step by step but become too expensive because they need many powerful LLM calls, and (2) Retrieve-on-Graph methods that are cheaper but rely too much on retrieval and can miss key nodes/edges. Inspired by dual-process cognition (fast retrieval + slow reasoning) and information-theoretic tradeoffs, the authors propose Re³, a three-stage KGQA framework: (i) a fast, cognitively inspired retrieval to get a broad but noisy subgraph, (ii) a path-aware reranking step using a lightweight cross-encoder to prune and organize the retrieved paths into coherent reasoning chains, and (iii) an agentic reasoning stage where an LLM reasons over this high-quality subgraph and calls KG tools only when needed. On WebQSP and CWQ, Re³ is reported to outperform prior methods on complex multi-hop questions while keeping LLM calls under control.

**Strengths:**

1.The paper doesn’t just say “retrieve then reason,” it ties the design to cognitive science (fast vs slow thinking) and information theory (coverage vs capacity), which gives a clearer rationale than many ad-hoc KGQA pipelines.

2.Splitting into retrieval → path-aware rerank → agentic reasoning is practical: each stage can be swapped or scaled depending on resources (smaller reranker, cheaper LLM, different retriever).

3.Empirical validation on multi-hop benchmarks show gains on WebQSP and CWQ—datasets where incomplete retrieval really hurts—supports the claim that better path selection actually helps LLM reasoning, not just retrieval metrics.

**Weaknesses:**

1.the proposed method incorporates a more complicated pipeline for KGQA tasks, with three stages and additional models like the ranker. As a system, this work is good and also exhibits better performance. But it is hard to say whether the technical contributions of this paper are also insightful and useful for other domains. Actually, it is more promising to build the complex system for more complicated tasks like complex agent reasoning or real-world planning tasks. More comparison with other system-level designs in more complex tasks should be added in this paper.

2.Important baselines are not discussed or compared in this paper, e.g., KG-Agent, Chain-of-query

**Questions:**

Please refer to the weakness part. I agree this is a good system paper, but the technical novelty and contribution are not very clear.

**Details Of Ethics Concerns:**

Please refer to the weakness part.

---

> ### Author Response · Authors · 2025-12-03
> **Clarification on Modularity, Generalizability, and Task Scope**
>
> We thank the reviewer for recognizing that our system "exhibits better performance" and is a "good work." We appreciate the suggestion regarding the applicability of our framework to broader domains. We would like to address the concerns regarding complexity and technical contribution as follows:
>
> ---
>
> 1) **Modularity and Flexibility vs. Complexity**
>
> While Re$^{3}$ involves three stages, we respectfully argue that this design represents a modular and decoupled architecture rather than a "complicated pipeline."
>
> First, unlike methods that rely on heavy, task-specific fine-tuning of Large Language Models (LLMs), our components (e.g., KGE embeddings, standard Cross-Encoders, and off-the-shelf LLMs) are general-purpose. As noted in Section 3, each module is "plug-and-play."
>
>
> And, a key advantage of this pipeline over end-to-end "black box" models is controllability. By explicitly separating Retrieval (context acquisition), Reranking (information filtering), and Reasoning (answer derivation), we gain fine-grained control over the Recall-Precision trade-off, which is difficult to achieve in monolithic systems.
>
>
> Last, as mentioned in our "Efficiency Analysis" (Section 4.6 ), the inclusion of the Reranking module actually reduces the overall computational burden by filtering noise before it reaches the computationally expensive LLM agent.
>
> ---
>
> 2) **Theoretical Insight and Transferability**
>
> We believe the technical contributions of Re$^{3}$ offer insights applicable to many domains beyond KGQA, specifically for Retrieval-Augmented Agents:
>
>
> (1) Our framework is grounded in the dual-process theory (Fast vs. Slow thinking). The architectural decision to use a fast, high-recall retriever followed by a slower, deliberate agentic reasoner is a general design pattern suitable for any knowledge-intensive task (e.g., legal analysis, medical diagnosis), not just KGQA.
>
>
> (2) The mechanism where the agent dynamically decides to invoke tools only when information is insufficient (Section 3.3 ) is a generic solution to the "Efficiency-Accuracy" dilemma in agentic planning.
>
> ---
>
> 3) **Comparison on Complex Tasks**
>
> We agree with the reviewer that this architecture holds great promise for "complex agent reasoning or real-world planning tasks." However, we selected Multi-hop KGQA (specifically WebQSP and CWQ) as our testbed because it provides a rigorous, structured environment to quantitatively evaluate the system's ability to handle long-horizon reasoning and noise filtering.
>
> KGQA is widely recognized as a proxy for complex reasoning capabilities.
>
> While extending Re$^{3}$ to open-ended planning tasks is a compelling direction for future work, the current scope focuses on solving the critical "Recall-Precision" and "Cost-Quality" trade-offs  inherent in structured knowledge reasoning. We believe our extensive comparisons against strong baselines like ToG and RoG  sufficiently demonstrate the system-level superiority in this context.
>
> ---
>
> 4) **Concern about the baselines**
>
> We acknowledge these are relevant works. However, we excluded them to ensure a fair and reproducible comparison, following recent community standards:
>
> KG-Agent: Its official implementation is not publicly available. Consequently, recent state-of-the-art studies (e.g., EPERM [AAAI 2025], FRAG [ACL 2025], GCR [2024]) have also excluded it from their baselines. We followed this consensus to prioritize verifiable methods.
>
> - Luo, L., Zhao, Z., Haffari, G., Li, Y. F., Gong, C., & Pan, S. (2024). Graph-constrained reasoning: Faithful reasoning on knowledge graphs with large language models. arXiv preprint arXiv:2410.13080. ICML
>
> - Long, X., Zhuang, L., Li, A., Yao, M., & Wang, S. (2025, April). Eperm: An evidence path enhanced reasoning model for knowledge graph question and answering. In Proceedings of the AAAI Conference on Artificial Intelligence (Vol. 39, No. 12, pp. 12282-12290).
>
> - Li, M., Miao, S., & Li, P. (2024). Simple is effective: The roles of graphs and large language models in knowledge-graph-based retrieval-augmented generation. arXiv preprint arXiv:2410.20724.
>
> - Gao, Z., Cao, Y., Wang, H., Ke, A., Feng, Y., Zhou, S. K., & Xie, X. (2025, July). Frag: A flexible modular framework for retrieval-augmented generation based on knowledge graphs. In Findings of the Association for Computational Linguistics: ACL 2025 (pp. 6178-6192).
>
> Chain-of-Query: It does not report results on the standard WebQSP and CWQ benchmarks used in our study (and widely in the field). Direct comparison is not feasible due to misaligned experimental settings.
>
> Our comparison focuses on widely recognized, open-source baselines (e.g., ToG, RoG, SubgraphRAG) to ensure the reliability of our conclusions.
>
> ---
>
> We hope this clarifies that Re$^{3}$ is designed as a paradigm for efficient agentic reasoning, balancing system complexity with significant gains in interpretability and performance.

---

### Note · Program_Chairs · 2026-01-17
**Submission Desk Rejected by Program Chairs**

The following references in this submission do not refer to real documents and/or have major errors in bibliographic information:

 Yongqiang Feng, Xien Liu, Lehan Qu, Zequn Zhang, Mingxiang Chen, Xin-Qiang Cai, Ming-ming Yang, Wenge Rong, and Zhang Xiong. Subgraph-based retrieval-augmented generation framework for knowledge graph question answering. In Proceedings of the 2024 Joint International Conference on Computational Linguistics, Language Resources and Evaluation, 2024. URL https://aclanthology.org/2024.lrec-main.502/.